# Active ballistic orbital transport in Ni/Pt heterostructure

Sobhan Subhra Mishra[1,2], James Lourembam [3], Dennis Jing Xiong Lin [3] & Ranjan Singh [1,2] ✉

Orbital current, defined as the orbital character of Bloch states in solids, can travel with larger coherence length through a broader range of materials than its spin counterpart, facilitating a robust, higher density and energy efficient information transmission. Hence, active control of orbital transport plays a pivotal role in the progress of the evolving field of quantum information technology. Unlike spin angular momentum, orbital angular momentum couples to phonon angular momentum efficiently via orbital-crystal momentum (L-k) coupling, allowing us to control orbital transport through crystal field potential mediated angular momentum transfer. Here, leveraging the orbital dependant efficient L-k coupling, we have experimentally demonstrated the active control of orbital current velocity in Ni/Pt heterostructure. We observe terahertz emission from Ni/Pt heterostructure via long-range ballistic orbital transport, as evidenced by the delay, and chirping in the emitted THz pulse correlating with increased Pt thickness. Additionally, we also have identified a critical energy density required to overcome collisions in orbital transport, enabling a swifter flow of orbital current. Femtosecond light driven active control of the ballistic orbital transport lays the foundation for the development of dynamic optorbitronics for transmitting information over extended distance.

The interconversion of spin current and charge current has been extensively explored in the field of spintronics research[1,2]. Notable instances of this conversion encompass phenomena such as spin Hall[2] and inverse spin Hall effect[3,4] observed in heavy metals, spin momentum locking in topological insulators[5,6], and the Rashba-Edelstein effect along with its inverse counterpart in two-dimensional electron gases and other systems having broken inversion symmetry[7–9]. In recent times, these effects have garnered substantial traction as potent methods for harnessing the spin angular momentum carried by electrons[10,11], allowing for the generation of ultrafast charge current, thereby enabling the emission of broadband THz pulse[12,13].

Recent research has brought to light the significance of the orbital angular momentum of electrons, giving rise to the emerging field of orbitronics[14,15]. It can possess substantial technological advantage over its spin counterpart[16]. Additionally, the conversion between orbital and charge current does not require a heavy metal[17,18], thereby expanding the range to include light metals available for utilization. One of the limitations of these systems is the absence of a direct source of orbital current; however, this limitation can be eliminated by various orbital pumping techniques through magnetization and lattice dynamics. Orbital Angular Momentum pumping through magnetization dynamics can be explained by transfer of spin angular momentum to orbital angular momentum through Spin-orbit Coupling (SoC) in

[1]Division of Physics and Applied Physics, School of Physical and Mathematical Sciences, Nanyang Technological University, Singapore 637371, Singapore. [2]Centre for Disruptive Photonic Technologies, The Photonics Institute, Nanyang Technological University, Singapore 639798, Singapore. [3]Institute of Materials Research and Engineering, Agency for Science, Technology and Research, 2 Fusionopolis Way, Singapore 138364, Singapore. ✉e-mail: ranjans@ntu.edu.sg

ferromagnet[19–21]. This can be achieved through ultrafast photoexcitation of a high SoC ferromagnet (like Ni) which results in generation of spin current. The spin current can be converted to orbital current mediated by spin-orbit interaction in the ferromagnet, making it an indirect source of orbital current. Another technique of orbital pumping is through lattice dynamics via efficient orbital-dependent electron-phonon coupling[22]. This can be explained as a reverse process of crystal field torque[23]. The orbital current then transports to the adjacent nonmagnetic (NM) metal layer, where it converts to an accelerated charge current generating THz waves as shown in Fig. 1a[24,25]. Recent research has proposed that orbital transport can be ballistic and can propagate over long distance, reaching up to 80 nm in some of the widely used nonmagnetic layers like Tungsten (W) at a velocity measured up to 0.14 nm/fs[25]. Various other works have also described a long-range transport of orbital current in several metals[26–28] including via THz emission[29].

Here, we propose an active method to control the orbital current velocity through the applied optical fluence in a Ni/Pt heterostructure. Our findings demonstrate that upon ultrafast photoexcitation, the Ni/Pt heterostructure emits THz radiation primarily due to long-range ballistic transport of orbital current within the Pt layer, which is subsequently converted to charge current. Harnessing electron-phonon coupling which is dependent on orbital angular momentum but independent of spin, we can actively control the orbital transport through laser fluence. Our findings demonstrate tunable orbital current velocities ranging from 0.14 nm/fs to 0.18 nm/fs, exhibiting a direct control over the velocity of orbital transport. We also determine the critical energy density necessary to overcome collisions, thereby swifter movement of orbital current within the metal layer is achieved.

## Results and discussion

Spin and orbital angular momentum share similar symmetry and magnetization dynamics. However, they exhibit distinct behaviors on ultrafast timescale[25,30,31]. Upon ultrafast photoexcitation in

ferromagnet layer, ultrafast demagnetization occurs, creating a difference between equilibrium and instantaneous magnetization, resulting in spin accumulation and generation of corresponding spin current[32,33]. Through similar magnetization dynamics, an orbital accumulation may arise facilitated by SoC in the ferromagnet generating an orbital current. Despite the similarity in magnetization dynamics[19,34], the transport of both spin and orbital currents within the adjacent nonmagnetic layer can vary with its thickness, as their transport properties differ at interfaces and within the bulk of NM materials. Additionally, due to applied electric field of light, the angular momentum exchange between the lattice and orbital wave function can also control the transport of orbital angular momentum[22].

As depicted in Fig. 1a, when the FM/NM heterostructure is subjected to ultrafast photoexcitation, the initially generated ultrafast spin current partially transforms into an ultrafast orbital current due to L.S correlation in ferromagnet layer, resulting in the injection of both spin ($j_S$) and orbital current ($j_L$) into the nonmagnetic metal layer. Through the processes of LCC and SCC, an accelerated charge current is induced, serving as the source of THz radiation. The resultant THz electric field is directly proportional to the transient sheet charge current ($I_C(t)$), as described by the following equation[12,13,35].

$$\mathbf{E}(t) \propto I_C(t) \propto \int_{-d_{FM}}^{d_{NM}} dz \left[ \theta_{LC}(z) j_L(z,t) + \theta_{SC}(z) j_S(z,t) \right] \quad (1)$$

$d_{FM}$ and $d_{NM}$ denote the thickness of the ferromagnet and nonmagnetic metal layer respectively and $\theta_{LC}$ and $\theta_{SC}$ denote the orbital Hall and spin Hall angle, which governs the efficiency of orbital to charge (LCC) and spin to charge conversion (SCC). The equation does not include the minor process of THz emission like ultrafast demagnetization and anomalous hall effect due to the ferromagnet as those can be separated out from the photocurrent mechanisms experimentally[36]. To isolate magnetic effect, we have considered the signals arising from the difference in the signals when the magnetic

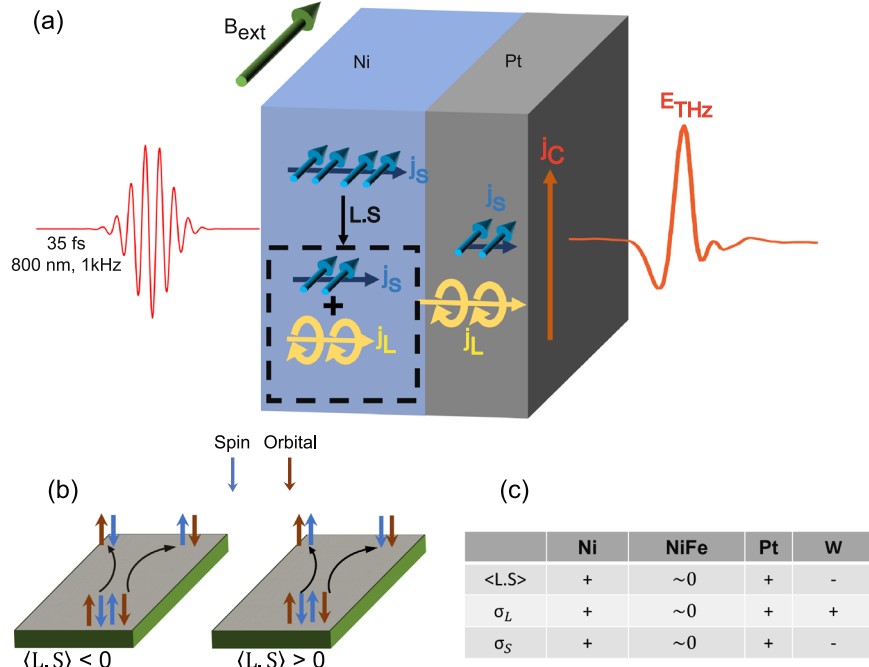

**Fig. 1 | Pumping and detection of terahertz orbital (L) and spin (S) currents. a** Upon ultrafast photoexcitation of the FM, spin currents are generated and partially converted to orbital currents due to L.S correlation, injecting both spin current ($j_S$) having shorter relaxation length and orbital current ($j_L$) having longer relaxation length into metal layer. Orbital-to-charge conversion (LCC) and spin-to-charge conversion (SCC) processes generate an accelerated charge current ($j_C$) to emit a THz radiation. **b** L.S correlation dependence of direction of orbital current and spin current propagation in Orbital Hall effect. **c** Sign of spin-orbit coupling and the orbital ($\sigma_L$) and spin ($\sigma_S$) hall conductivity for Ni, NiFe, Pt, and W.

field is reversed ($[E_{THz}(+M) - E_{THz}(-M)]/2$). The signal due to anomalous Hall Effect in Ni is nearly insignificant compared to the emitted signal from Ni/Pt in the thickness regime of the Ni layer investigated.

Relative sign of spin hall and orbital hall angle depends on the L.S correlation ⟨L.S⟩ as shown in Figs. 1b, c. If L.S < 0, the spin and orbital transport will have opposite sign whereas if L.S > 0, spin and orbital transport will have same sign[37]. To facilitate a direct comparison between orbital transport and spin transport, we study two different types of FM/NM heterostructures. In the first case, Ni was chosen as the FM layer to illustrate orbital transport as Ni has higher efficiency of generating orbital current from spin current because of its higher L.S correlation[38]. In the second case, to demonstrate spin transport, NiFe is chosen as FM due to its higher spin current generation efficiency[12]. In both the cases, NM chosen was Pt due to its high Orbital Hall Conductivity and Spin Hall Conductivity[39]. An in-plane constant magnetic field of 128 mT was applied to keep the system at saturated magnetization state.

Figure 2a displays the emitted THz radiation from a Ni (3 nm)/Pt ($x$ nm) heterostructure, when photoexcited by a constant fluence of 1270 µJ/cm$^2$, with variable Pt thickness ($x$ = 3, 6, 9, 18 nm). It is observed that as the Pt thickness increases, the emitted THz pulse encounters a delayed arrival, whereas in case of NiFe (3 nm)/Pt ($x$ nm) with $x$ = 1, 2, 4, 6, 8, there is no delay in THz pulse as illustrated in Fig. 2b. This comparison serves to establish the prevalence of different photocurrent mechanism of THz emission from Ni/Pt compared to NiFe/Pt. In case of NiFe/Pt where spin transport dominates the THz emission, the increase in Pt thickness does not delay the arrival of THz due to lower relaxation length of spin current in Pt[40,41]. It is worth noting that the spin transport can be ballistic and long-range order in other material systems and interfaces[42,43], however it is well known that the spin diffusion length of Pt is around 1.2 nm[40,41]. A very small undetected delay of around 5-6 fs

can be induced (See supplementary section S3 for calculations) due to the THz refractive index of platinum. Additionally previous experiment suggests that even when Pt thickness is increased upto 20 nm, no shift in THz pulse is observed in case of Co /Pt heterostructure[44]. Similar behavior was also observed in case of Ni (3 nm)/Ru ($x$ nm) where $x$ is ranged from 3 to 18 (See supplementary section S10). There is a considerable shift in the emitted pulse upto 6 nm of Ru thickness. However, as we increase the thickness further, the delay is not observable, indicating a relaxation length of <6 nm of Ru, which has been verified experimentally in previous literature[45]. Therefore, the delay in Ni/Pt could be attributed to a long-distance transport in Pt thus ruling out the spin transport as the spin diffusion length of Pt is shorter (-1.2 nm)[40,41] proving that the THz emission is due to the long-range transport thus indicating an orbital transport mechanism in Ni /Pt heterostructure.

The shift in the peak of the emitted THz pulse is graphically represented in Fig. 2c, demonstrating the linear correlation between Pt thickness till 18 nm and the delay obtained in THz peak, which validates the ballistic nature of observed long-range orbital transport in Pt till at least 18 nm and provides us with an orbital current velocity of approximately 0.18 nm/fs, as opposed to the zero delay in THz peak value when increasing the thickness of Pt in the case of NiFe/Pt, where spin transport predominates. Furthermore, due to its larger relaxation length in Pt (Please see the supplementary section S4 for further explanation), orbital current will experience more significant angular dispersion at higher thicknesses, resulting in the chirping of the THz pulse depicted in Fig. 2d. A linear decay of the THz signal normalized to pump fluence at different thickness of Pt (Supplementary Section S4, Figure S3) gives a strong indication of long-range transport mechanism. Additionally, THz pulse chirping is also observed at larger thickness of Pt in case of Ni/Pt as seen in Fig. 2d. However, in case of NiFe/Pt where spin transport dominates, due to very short relaxation length of

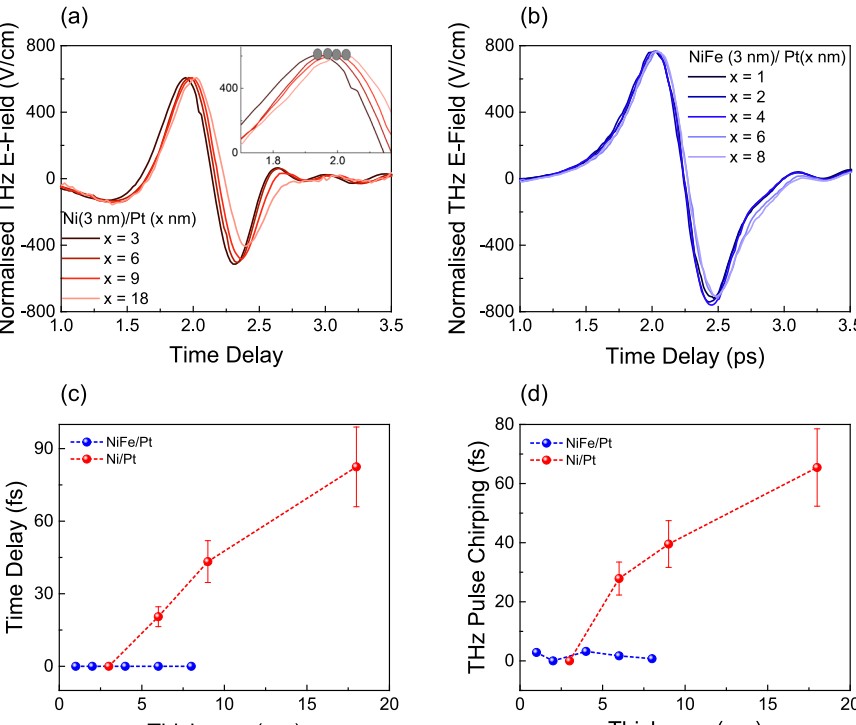

**Fig. 2 | Ballistic orbital transport in Ni/Pt heterostructure. a** Terahertz signal emitted from Ni(3 nm) /Pt($x$ nm) heterostructure with variable Pt thickness $x$ = 3, 6, 9, and 18; Inset shows the zoomed-in peak of the THz pulse indicating a right shift as we increase the Pt thickness. **b** Terahertz signal emitted from NiFe (3 nm) /Pt ($x$ nm) heterostructure with variable Pt thickness with $x$ = 1, 2, 4, 6, and 8 indicating no right shift in the peak. **c** Linear delay in the emitted THz pulse with an increase in Pt

thickness demonstrating the ballistic orbital transport in Ni/Pt heterostructure in contrast to spin transport in NiFe/Pt heterostructure. **d** Chirping of the emitted THz pulse with an increase in Pt thickness in Ni/Pt heterostructure providing evidence of the increase in the angular dispersion of the transport entity, proving the longer relaxation length of orbital current in contrast to spin transport in NiFe/Pt heterostructure.

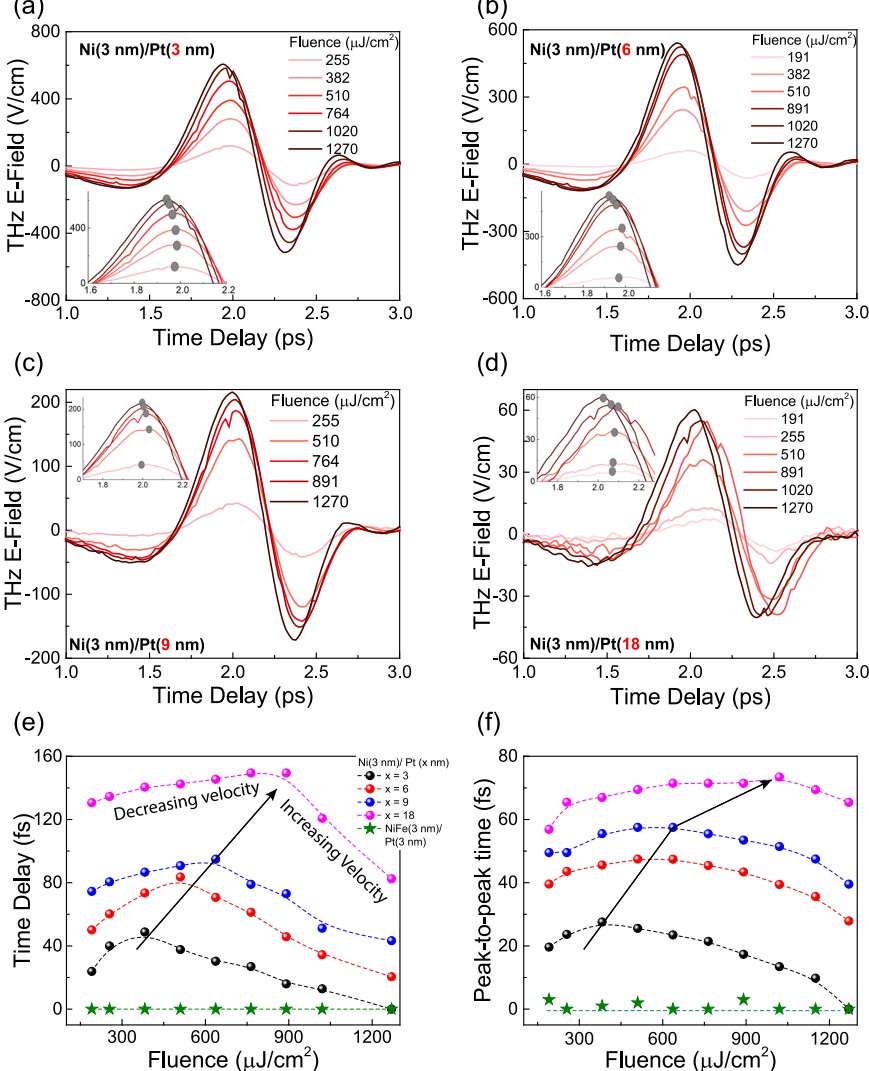

**Fig. 3 | Light-driven active control of ballistic orbital transport and critical laser fluence.** THz emission from Ni (3 nm)/Pt (x nm) at different fluence when **a** x = 3, **b** x = 6, **c** x = 9, **d** x = 18, A clear shift in THz peak was observed as we change the fluence. **e** Extracted time delay for different fluence is shown for Ni (3 nm)/Pt (x nm) with x = 3, 6, 9, 18. Initially, the shift is towards right indicating the decrease in orbital velocity and after a critical fluence, delay starts decreasing with increase in fluence showing swifter orbital transport; Similar shift is not seen in spin transport as shown in NiFe (3 nm)/Pt (3 nm). **f** THz pulse chirping, for different fluence is shown for Ni (3 nm)/Pt (x nm) with x = 3, 6, 9, and 18. A thickness-dependent critical fluence similar to **e** can be seen.

spin in Pt[40,41], the spin current does not travel enough distance to have large angular dispersion resulting in the absence of THz pulse chirping with an increase in thickness of Pt. A gradual chirping of THz pulse in case of Ni/Pt also eliminates the additional factors such as strain, higher defect density or other physical properties. Furthermore, the existence of these minor factors will only be reflected in the emitted THz pulse in the thickness regime investigated if it is accompanied by a long-range transport phenomenon ruling out the short-range spin transport in Pt with a relaxation length of around 1.2 nm[40,41], resulting in the absence of pulse chirping as seen in case NiFe/Pt (x nm). This further confirms that a long-range transport mechanism is responsible for THz emission in Ni/Pt heterostructure, excluding the possibility of short-range spin transport[40,41] thereby strongly indicating orbital transport.

The orbital angular momentum of the nonlocalized electrons interacts with the lattice through the crystal field potential with following continuity equation[23,46].

$$\frac{\partial}{\partial t}\langle L \rangle = \langle F^L \rangle + \frac{1}{i\hbar}\langle [L, V_{CF}] \rangle + \langle \lambda S \times L \rangle \qquad (2)$$

Here $F^L$ is the orbital flux term, $\langle [L, V_{CF}] \rangle$ describes the transfer of angular momentum between orbital and crystal, $V_{CF}$ is crystal field potential and $\lambda S \times L$ describes the mutual transfer of spin and orbital angular momentum within a single electron. The electric field of the applied laser fluence interacts with the orbital, causing a perturbation in the orbital wave function, which gives rise to a non-equilibrium orbital wave function. The electric field perturbed non-equilibrium orbital wave function extracts angular momentum from the lattice, thus affecting the interaction of the orbital angular momentum and the crystal field potential, thereby increasing the velocity of orbital transport.

Figure 3 offers a comprehensive illustration of the active control of observed orbital transport within the Pt layer. The thickness of the ferromagnetic material Ni remains constant at 3 nm throughout the experiment. In Fig. 3a–d, the emitted THz pulse is depicted for Ni (3 nm) and Pt (x nm), where x takes values of 3, 6, 9, and 18, respectively, at different fluence levels. As the fluence increases, a noticeable shift in the emitted THz pulse is observed. Initially, the pulse shifts towards the right, indicating a delayed arrival of the THz pulse. However, beyond a certain fluence threshold referred to as the *critical*

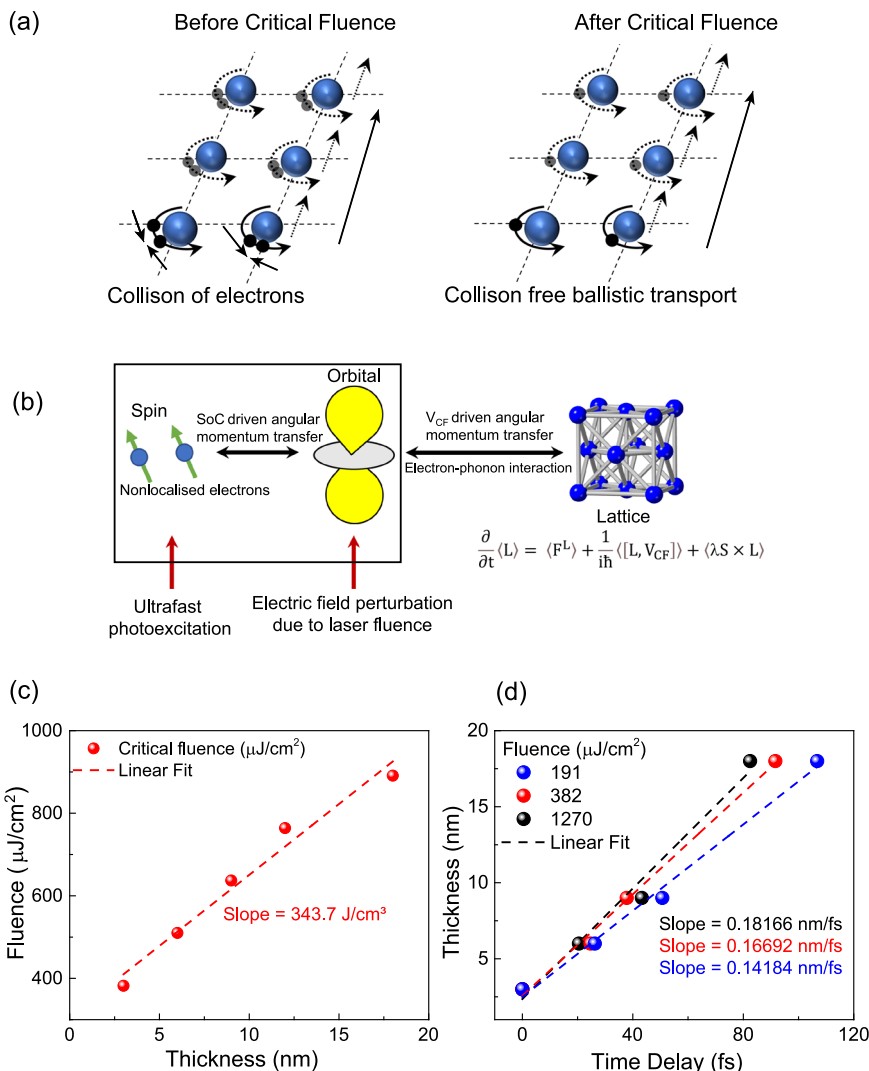

**Fig. 4 | Active control of orbital current velocity. a** Schematic of transport of orbital angular momentum, before critical fluence, collision between charge carriers make the transport slower, after critical fluence, swifter transport of orbital angular momentum takes place. **b** Mechanism of increase in orbital current velocity due to angular momentum transfer between lattice and electrons. **c** Extracted critical fluence with variation in thickness. The slope of the fitted straight line indicates the "critical energy density". **d** Extracted time delay with thickness at different fluences. The slope of the linear fitted line can be termed as orbital current velocity which could be tuned from 0.14 nm/fs to 0.18 nm/fs as the fluence was increased from 191 µJ/cm² from 1270 µJ/cm².

*fluence*, there is an increase in orbital current velocity. Consequently, beyond this *critical fluence*, there is left shift in the arrival time of the THz pulse. Similar behavior was also observed in other orbital transport-based emission system Ni/Ru (See supplementary section S10). Figure 3e demonstrates that critical fluence is contingent on the thickness of the nonmagnetic Pt layer. As the thickness is increased, the carriers require more energy to overcome collisions to move ballistically over a larger distance, resulting in a higher critical fluence. Similar behavior was also observed when the peak-to-peak time was recorded as shown in Fig. 3f.

Figure 4a elucidates the schematic representation of the orbital transport mechanism both prior to and post reaching critical fluence. Nonlocalized electrons, bearing information about orbital angular momentum, traverse through orbital hopping between nuclei within a solid. An increase in fluence increases the number of charge carriers, thereby enhancing the number of collisions that impede the transport process. Nevertheless, beyond the critical fluence, nonlocalized electrons perturbed by laser fluence absorb additional angular momentum from the lattice according to Eq. 2[22,23,46], as illustrated in Fig. 4b. As we apply laser fluence, the local magnetic moment couples with the spin

of the nonlocalized conduction electrons through exchange interaction[10,23], thus creating a spin current. Due to high spin-orbit correlation of Ni near Fermi level, there is angular momentum transfer between spin and orbital which can be explained by the cross product of S and L ($\langle \lambda S \times L \rangle$) as shown in Eq. 2. Additionally, due to the electric field of the applied laser fluence, a perturbation in the orbital wave function is induced, thus creating a non-equilibrium state. Consequently, the non-equilibrium orbital wave function extracts angular momentum from the lattice through the crystal field potential $V_{CF}$ explained by $\frac{1}{i\hbar}\langle [L, V_{CF}] \rangle$ in the Eq. 2, enabling the orbital current to surpass collisions, facilitating a more rapid ballistic transport.

Clearly, the critical fluence exhibits a linear correlation with the thickness of the heavy metal layer. Figure 4c illustrates that as the thickness increases, the critical fluence also increases proportionally. The slope of the linear relation can be termed as *critical energy density* denoted by $\varepsilon_C$ and quantified as 343.7 J/cm³ in case of Ni/Pt heterostructure representing the energy required per unit volume to overcome the collision and facilitate a complete ballistic orbital transport in Pt. For different material systems, $\varepsilon_C$ can be different depending on the intrinsic properties of the materials, and further investigation is

required. Finally, the orbital current velocities at different fluences were extracted by recording the delay in the THz peak with respect to the Ni (3 nm)/Pt (3 nm) heterostructure. Given the ballistic nature of orbital transport, the slope of the linear relationship between delay and the heavy metal thickness was utilized to calculate the velocity. As shown in Fig. 4d, the orbital current velocity was extracted at 191 μJ/cm² and found out to be 0.14 nm/fs. As we increased the fluence, the slope started increasing and at 382 μJ/cm² the orbital current velocity was 0.16 nm/fs and at 1270 μJ/cm², the velocity was found to be 0.18 nm/fs. The linear relation between the delay in the THz pulse and the thickness of Pt also indicates the ballistic nature of the orbital transport in Pt layer over a larger distance than spin transport.

In summary, using THz emission spectroscopy, we observed that the predominant source of THz emission from femtosecond light-excited Ni/Pt heterostructures is long-range ballistic orbital transport. Furthermore, the orbital transport can be controlled through the electric field of the incident femtosecond laser pulses. Absorption of higher energy initially leads to more charge carrier formation enhancing the collision, thus delaying the transport. However, after a critical fluence of the femtosecond laser beam, carriers overcome the collision, and the orbital transport occurs at a swifter pace due to transfer of phonon angular momentum. Exploiting this phenomenon, we have also highlighted the active enhancement of experimentally calculated orbital current velocity from 0.14 nm/fs to 0.18 nm/fs through increase in fluence enabling faster orbital transport. Our findings establish an approach to control the long-distance ballistic L transport, thus creating new opportunities to design future ultrafast devices with optorbitronic materials. Additionally, because of the orbital-dependent efficient phonon orbital coupling, it is also possible to have lattice-assisted orbital pumping known as Orbital Angular Position (OAP)[22,47]. By, integrating optics, and orbitronics towards THz emission, a magnetic field free Optorbitronic Terahertz Emitters (OTE) would be of immediate practical importance.

## Methods

### Sample preparation
The FM/NM films were deposited on 1mm- Quartz substrates by d.c. magnetron sputtering at room temperature, using a Chiron ultrahigh vacuum system with a vacuum base pressure of $1 \times 10^{-8}$ torr.

### THz emission experiment
The THz radiation emitted is captured using a 1 mm thick ZnTe crystal oriented along the <110> axis, known for its nonlinear properties. The femtosecond laser pulse, which illuminates the orbitronic hetero-structure, has a wavelength of 800 nm, corresponding to a laser energy of 1.55 electron volts (eV). It has a pulse width of 35 femtoseconds (fs) and operates at a repetition rate of 1 kHz. A beam splitter is employed to divide it into two parts. The higher intensity laser beam is directed towards photoexcitation of the emitter, while the lower intensity beam serves as a probe for detection. Precise time matching is ensured by a mechanical delay stage. The step size of the delay stage is 3 μm, which corresponds to a time difference of 20 fs between the data points in THz pulse, enabling a precise extraction of the delay in the peak of the emitted THz. The details about the THz emission spectroscopy set up can be found in the supplementary section S1. As the emitted THz pulse, collected by parabolic mirrors, focuses on the ZnTe <110> detector, it induces birefringence in the ZnTe detector crystal. Simultaneously, the time-matched probe beam traverses through the crystal and encounters a change in its polarization, directly proportional to the birefringence. For the electro-optic detection[48], a quarter-wave plate and a Wollaston prism are employed to distinguish the s and p polarization of the probe beam. The changes in the probe laser are subsequently identified using a balanced photodiode that measures the intensity difference between the s and p -polarized light. The electrical signal thus detected undergoes initial pre-amplification before being input into the lock-in amplifier to enhance the signal-to-noise ratio. The resultant signal from the lock-in amplifier is utilized to plot the THz electric field.

### Reporting summary
Further information on research design is available in the Nature Portfolio Reporting Summary linked to this article.

## Data availability
The Figure data generated in this study have been deposited in the DR-NTU database at https://doi.org/10.21979/N9/FUGSDM. All the other data are available from the corresponding author upon reasonable request.

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

## Acknowledgements

The authors thank Thomas Tan and Baolong Zhang for valuable discussions and suggestions. R.S. and S.M. would like to acknowledge the Ministry of Education (MoE), Singapore, for the support through MOE-T2EP50121-0009. J.L. and D.J.X.L would like to acknowledge funding support from SpOT-LITE programme (Agency for Science, Technology and Research, A*STAR Grant No. A18A6b0057) through RIE2020 funds from Singapore.

## Author contributions

S.M., J.L., and R.S. conceived the project. S.M. and R.S. designed the experiments for light driven active control of orbital current velocity. S.M. performed all the THz emission measurements and experimental analysis. D.J.X.L. and J.L. fabricated the orbitronic emitters. All the authors analyzed and discussed the results. S.M. and R.S. wrote the manuscript with input from J.L. R.S. led the overall project.

## Competing interests

The authors declare no competing interests.
