## [Peer Review File · Nature Communications]

Reviewers' comments:

Reviewer #1 (Remarks to the Author):

Review report on “Active Control of Ballistic Orbital Transport” by Mishra et al., submitted to Nature Communications.

In my opinion, the manuscript cannot be published in a journal because the THz experiments on Ni/Pt and NiFe/Pt as well as the interpretations used in it have been borrowed from previously published scholarly work which has provided rather much more extensive experimental data on those facts. Not only the duplication of the THz experiments and major assertions but also some texts have been borrowed as is from the previous literature. Vast similarities are with the article by Seifert et al., Nature Nanotechnology 18, 1132–1138 (2023), except for the fact that the authors in this reference paper have correctly used the nonmagnetic layer interfaced with a SiO₂ layer to demonstrate ballistic orbital transport and IOREE induced orbital to charge conversion responsible for the THz emission. Please see Supplementary information also of Seifert et al., Nature Nanotechnology 18, 1132–1138 (2023), for a comparison where the experimental results with respect to fluence are different for the same type of bilayer. In addition, some of the previous works on similar lines have not been appropriately credited. See, Choi et al., Nature 619, 52–56 (2023); Kumar et al., Nature Communications 14, 8185 (2023); Seifert et al., Nature Nanotechnology 18, 1132–1138 (2023); Wang et al., npj Quantum Materials 8, 28 (2023); Go et al., Physical Review Letters 130, 246701, (2023), etc.

Only to help the editorial evaluation of the current manuscript, I have provided a few points below to flag the issues.

Except for the degree of spin polarization that is controlled by alloy composition in NiFe, it's bilayers with any nonmagnetic layer have been shown to exhibit similar behaviors in many ultrafast THz experiments reported earlier. The main results of the current manuscript are in Figs. 2 and 3. The claim that the emitted THz from Ni/Pt is primarily due to orbital transport is not well supported, instead, the same can be well explained solely by spin transport within the purview of the data presented in this manuscript. Ultrafast excitation of Ni in Ni/Pt generates spin current and a fraction of spin current (J_S) is converted into orbital current (J_L) (in consistency with the L.S conversion in Ni, $J_L \ll J_S$). Since the orbital and spin Hall conductivities are almost the same in Pt, the spin to charge conversion is way higher. In absence of any subsequent layer after Pt, the spin to orbital conversion in Pt due to large spin-orbital conversion in it goes as waste. THz emission from Ni/Pt is simply due to ISHE in which larger Pt layer thickness delays, attenuates and broadens the THz pulse. The spin and orbital diffusion lengths in Pt should also be properly discussed and used.

Figure 2: Data presentation for Ni(3)/Pt(x) where x varies from 3 to 18 nm and NiFe(3)/Pt(x) where x varies from 1 to just 8 nm is misleading and hence the comparisons in the behaviors in subparts (c) and (d) are erroneous. To me, the behavior of both the bilayers is the same at this short time scale. Any apparent temporal shift (delay or broadening) in the orders of few fs to 10's of fs looks artificial owing to the fact that FWHM of the femtosecond pulses used to sample the THz pulses is ~50 fs or even larger at the sample point in such experiments. Moreover, such short temporal uncertainties are prone to arise from the variation in the thickness of the substrates used. What are the bidirectional repeatability and backlash error in the motorized delay stage used in the experiments? How was the laser fluence controlled and whether the delay for different OD's was properly considered? The apparent temporal

delays have been attributed to ballistic orbital transport in Ni/Pt as compared to spin transport in NiFe/Pt. However, if the delays are correct and such an inference is to be believed, then it rather suggests the opposite picture, i.e., ballistic spin transport due to no temporal shifts in NiFe/Pt.

Figure 3: Fluence dependent data for NiFe/Pt(x) for the apparent temporal shifts in the THz pulses with respect to x should also be presented to make a better comparison with the same for Ni/Pt(x). In my opinion, the results presented do not match the claims of the paper. The assumption that there is a critical value of the laser fluence to change the nature of the orbital transport, i.e. increasing or decreasing across this point by increasing the fluence, cannot be said unambiguously. The raw data has large experimental errors in the way the temporal shifts can be estimated exactly from them. With the increasing fluence, the THz signal from the spintronic heterostructures is believed to saturate. At the high fluence values used in the current paper, it will have a negative impact on the clear estimation of the temporal shifts.

Reviewer #2 (Remarks to the Author):

The authors report on an experiment where they pump nickel and permalloy layers with an adjacent platinum layer of varying thickness with a femtosecond laser of 800 nm wavelength. Afterwards, they collect the Terahertz radiation emitted by this system and probe its time delay. From the time delay measured, they draw conclusions on a new phenomenon, ballistic transport of electron current that carries an orbital momentum.

On the first glance, the paper reads well. The text is written in good English, the experiment is well described, and the measured data is shown. However, when diving deeper into the matter, the paper shows many deficits. I cannot recommend it for a publication in Nature Communications (and potentially also not elsewhere in the current state).

Whenever going away from the measured data, the background information, the discussion and conclusions read very general and superficial. There are a lot of claims wherever the proof or justification is not evident to the reader. I am going to discuss some of these statements in the following.

Starting from the abstract and the introduction, they are written in a very general manner. There are a lot of statements that are being made in a way I would expect clear explanations to the reader that are backed up with sound references. For instance, the authors claim that one can induce orbit currents into light metals using “various orbital pumping techniques”. What these techniques are remains undisclosed to the reader; the only reference given is a non-peer-reviewed article on arxiv.org. For such a central point to the study, I would expect both a good explanation why and how this works as well as one – ideally more – sound references. Further, the authors claim that spin transport is “super diffusive in nature”, giving one reference with a very specific material system and application case (ultrafast demagnetization). A brief web research yields a wealth of publications both for experimental proof of ballistic spin transport (<https://doi.org/10.1103/PhysRevB.65.155322>, both length of 150 nm and velocity of > 0.2 nm/fs exceeding the value of this publication), theory (<https://doi.org/10.1103/PhysRevB.96.115445>), or studies that even investigate when spin transport gets from a diffuse to ballistic regime (<https://doi.org/10.1103/PhysRevLett.124.196602>). I may assume that the authors believe that spin transport USUALLY occurs in a diffuse regime for their material system, but the statement here is certainly not correct.

Taking into account that the fundamental method to distinguish spin and orbit transport mechanisms in this study is the probing of the time delay under the assumption that a diffuse

transport mechanism does not lead to time delay variations when the platinum layer thickness is varied (by the way, shouldn't ballistic transport be faster than diffuse transport?), the whole central argumentation of the paper breaks down when one knows that spin transport can be ballistic. Is there any other way in the emission characteristics to determine whether the emitted Terahertz radiation has been created by a spin- or an orbit-transport mechanism? In any case, the orbital momentum must be conserved somehow. It would be possible to probe the emitted beam whether it carries an angular orbital momentum (under the assumption that the mechanism is as coherent as the authors claim in the model, which they describe in Figure 4).

Another uncertainty factor is that other effects must be excluded as the authors correctly reference to in line 138, reference 30. However, reference 30 describes that the ultrafast inverse spin-Hall effect (ISHE) easily dominates the emitted radiation and that the underlying effects must be disentangled with great care. However, I miss a clear and conclusive description how this was done during the experiment.

If we have a look to Figure 2d and the description that the widening of the peak is due to angular dispersion within the thickness, I ask myself whether this is the only possible explanation of a widening of that peak in the time domain. I can imagine a ton of other reasons, like strain, higher defect density or other physical properties that are varied with thickness that may influence the transport properties and widen the emission peak.

Another statement of the paper that worries me is the distinction of non-localized and localized electrons. I learned in fundamental physics that no electron is localized (says Heisenberg), and that orbitals give a probability of localization. Conduction bands in a metal can be seen as a large degenerate orbital (I'm fine with the description of a non-localized electron in this case), but what is the counterpart? Is it core electrons, or states that are in vicinity of a certain lattice location? The whole explanation remains diffuse at this point.

Apart from these fundamental concerns, I also have a couple of minor comments where the paper can be improved in technical details:

1. Figure 2c and d: I would swap the x and y axes.
2. There are no error bars.
3. Figure 3f: I cannot see the relation that is indicated by the arrow. I can see it in Figure 3e, but certainly not in Figure 3f (look, e.g., at the purple curve).
4. Caption to Figure 3 is incomplete.

Response Letter

Title: Active Control of Ballistic Orbital Transport

Manuscript # NCOMMS-24-01505-T, Nature Communications

The authors are thankful to the reviewers for their time and effort in examining our manuscript and providing constructive comments. We have carefully evaluated the constructive comments and concerns of the referees. They have been addressed with new experimental data, calculations, or clarifications, as relevant. These have led us to strengthen the evidence presented in support of our claims and add substantive novelty. In the text below, the reviewer's comment is followed by our detailed response, highlighted in blue. The modifications in the main manuscript as well as the supplementary information is indicated inside the textbox, highlighted in red.

Reviewer #1 (Remarks to the Author):

Review report on “Active Control of Ballistic Orbital Transport” by Mishra et al., submitted to Nature Communications.

In my opinion, the manuscript cannot be published in a journal because the THz experiments on Ni/Pt and NiFe/Pt as well as the interpretations used in it have been borrowed from previously published scholarly work which has provided rather much more extensive experimental data on those facts. Not only the duplication of the THz experiments and major assertions but also some texts have been borrowed as is from the previous literature. Vast similarities are with the article by Seifert et al., Nature Nanotechnology 18, 1132–1138 (2023), except for the fact that the authors in this reference paper have correctly used the nonmagnetic layer interfaced with a SiO₂ layer to demonstrate ballistic orbital transport and IOREE induced orbital to charge conversion responsible for the THz emission. Please see Supplementary information also of Seifert et al., Nature Nanotechnology 18, 1132–1138 (2023), for a comparison where the experimental results with respect to fluence are different for the same type of bilayer.

Response to the Reviewer: We appreciate the reviewer's thorough evaluation of our research. We acknowledge that our interpretation for calculating the orbital velocity draws inspiration from Seifert et al., a contribution duly cited in our reference 25 (21 in the original version).

However, we respectfully disagree with the assertion that certain passages have been directly “borrowed” from prior literature (Seifert et al. Nature Nanotechnology 18, 1132-1138 (2023)).

Our primary focus lies in the active manipulation of orbital velocity through the electric field induced by the applied laser fluence in the Ni/Pt heterostructure, as detailed in Figures 3 and 4 of our manuscript. We establish in Figure 2 that the THz emission originating from the Ni/Pt heterostructure is attributed to orbital transport, a novel observation not previously documented in the literature. This was validated using the methodology employed by Seifert et al¹ in their study of the Ni/W heterostructure, for which due credit and citation are provided in Reference 25 of our manuscript (Reference 21 in the original version). Furthermore, we highlight a related study by Xu et al. in Nature Communications, which presents a similar analysis within the context of Ni/Cu².

Considering the long range transport capability of orbital current within Pt, it can propagate through the thickness until its relaxation length within the nonmagnetic layer. We understand that in the presence of broken inversion symmetry, the orbital current has the potential to convert to charge current due to Inverse Orbital Rashba-Edelstein Effect (IOREE). Therefore, the conversion of orbital current to charge current at the Pt/air interface via IOREE is plausible without the need for an additional oxide layer.

The referee has also highlighted the potentially differing fluence-dependent results presented in the supplementary section of Seifert et al.’s work. Upon closer examination, it becomes apparent that the fluence range they explored is narrower compared to ours. Additionally, as we increase the fluence *THz pulse initially shifts to the right until reaching a critical fluence, after which it shifts back to the left* as depicted in Fig 3 of our main manuscript. Consequently, there exist certain fluence values where the THz pulses are closely spaced, resulting in imperceptible shifts. However, by reducing the step size in fluence change, we can observe the shifts in THz pulses. In Figure R1, we have adjusted the number of fluences depicted in Figure 3(a) of our main manuscript to match the ratio of fluences examined in Seifert et al.’s work. It is clearly seen that one can miss the potential shifts in THz pulses if the fluence step size is not chosen carefully.

Additionally, we also believe the difference in resolution and signal to noise ratio (SNR) in the experiment might be another reason of not observing the potential shifts.

Figure R1: Fluence dependent study of Ni (3 nm) / Pt (3 nm) (a) With all the fluences included indicating an initial shift of right and then left after a critical fluence in the THz pulse (b) With some of the fluences excluded giving an indication of no fluence dependent shift in the emitted pulse.

In addition, some of the previous works on similar lines have not been appropriately credited. See, Choi et al., Nature 619, 52–56 (2023); Kumar et al., Nature Communications 14, 8185 (2023); Seifert et al., Nature Nanotechnology 18, 1132–1138 (2023); Wang et al., npj Quantum Materials 8, 28 (2023); Go et al., Physical Review Letters 130, 246701, (2023), etc.

Response to the Reviewer: We appreciate the reviewer for bringing up prior works that align with our research. However, we kindly ask the reviewer to carefully examine our reference section. Both Seifert et al. (Nature Nanotechnology 18, 1132–1138, 2023) and Wang et al. (npj Quantum Materials 8, 28, 2023) are cited in our work, referenced as Reference 25 and Reference 24 respectively (21 and 20 in original version). It’s worth noting that the work by Kumar et al. (Nature Communications 14, 8185, 2023) was published after the submission of our manuscript to Nature Communications, but we have now appropriately cited their work (Reference 29 in the updated version).

We apologize for the oversight regarding the omission of Choi et al. (Nature 619, 52–56, 2023) and Go et al. (Physical Review Letters 130, 246701, 2023) in our initial manuscript. Both have been duly cited in the updated version of our manuscript (Reference 26 and 27 in updated version).

Only to help the editorial evaluation of the current manuscript, I have provided a few points below to flag the issues.

Except for the degree of spin polarization that is controlled by alloy composition in NiFe, it's bilayers with any nonmagnetic layer have been shown to exhibit similar behaviors in many ultrafast THz experiments reported earlier. The main results of the current manuscript are in Figs. 2 and 3. The claim that the emitted THz from Ni/Pt is primarily due to orbital transport is not well supported, instead, the same can be well explained solely by spin transport within the purview of the data presented in this manuscript. Ultrafast excitation of Ni in Ni/Pt generates spin current and a fraction of spin current (J_S) is converted into orbital current (J_L) (in consistency with the L.S conversion in Ni, $J_L \ll J_S$). Since the orbital and spin Hall conductivities are almost the same in Pt, the spin to charge conversion is way higher. In absence of any subsequent layer after Pt, the spin to orbital conversion in Pt due to large spin-orbital conversion in it goes as waste. THz emission from Ni/Pt is simply due to ISHE in which larger Pt layer thickness delays, attenuates and broadens the THz pulse. The spin and orbital diffusion lengths in Pt should also be properly discussed and used.

Response to the Reviewer: We thank the reviewer for raising this question. Our contention is that the observed shift in the THz pulse is not feasible with spin transport in Pt. Though spin transport can be long range and ballistic in some material systems^{3,4}, previous research has firmly established that the spin transport in Pt is short ranged and superdiffusive with spin diffusion length ranging between 1.2 nm and 1.6 nm in magnetic heterostructures^{5,6}. Thus, the observed delay in the emitted THz pulse, coupled with its chirping, indicates that the THz emission in the Ni/Pt heterostructure is attributable to the transport of an entity capable of traversing longer distances within Pt. Additionally if it were a purely spin transport mediated effect, the same delay should have been observed in NiFe / Pt (x nm) which is not the case. (Figure 2(b) in main manuscript and Figure R5 in the response letter).

Furthermore, potential factors such as thickness and refractive index of Pt contributing to the delay have been ruled out through THz transmission measurements, as detailed in our subsequent response (refer to Figure R7). This delay also underscores that the orbital diffusion length in Pt significantly surpasses its spin diffusion length. This observation can be further supported by examining the peak-to-peak THz amplitudes normalised to the absorbed fluence and Ni (3 nm)/ Pt (3 nm) for various Pt thickness, as illustrated in Figure R2.

Notably, the plotted data demonstrates a linear decrease in peak-to-peak THz amplitude at least until 18 nm of Pt thickness. When compared with transmitted pulse, it is observed that the emitted and transmitted peak to peak amplitude coincide at 18 nm suggesting the relaxation length to be greater than 18 nm in this system. We have added this in Supplementary Section S4.

Figure R2: THz emitted and transmitted peak to peak amplitude. The emitted pulse is normalized to the absorbed pump fluence. Both emitted and transmitted pulse is normalized to Ni (3) / Pt (3) sample. The emission pulse is decaying linearly whereas transmission pulse is decaying exponentially proving the long-range transport mediated THz emission.

The modification in the supplementary information is indicated below.

Section S4: Rough extraction of Orbital relaxation length

Figure S3: Comparison of THz transmission and THz emission for (a) Ni (3 nm) / Pt (x nm) showing a linear decay in the emitted THz pulse in contrast to an exponential decay in the transmitted THz pulse indicating a longer transport phenomena (b) NiFe (3 nm) / Pt (x nm) showing an exponential decay in both emitted and transmitted THz pulse indicating a shorter transport phenomenon.

It is anticipated that, beyond the relaxation length, both THz transmission and THz emission will display similar patterns with increasing thickness, coinciding with each other. Figure S3 depicts a comparison between THz transmission and emission for Ni/Pt and NiFe/Pt. In the instance of Ni/Pt (x nm), the peak-to-peak THz pulse emitted shows a linear decrease, indicating dispersion and attenuation, contrasting with the exponentially decreasing transmitted peak-to-peak THz pulse, until they coincide at a thickness of 18 nm, proving a relaxation length of greater than 18 nm. However, for NiFe/Pt, both the transmitted and emitted pulses exponentially decrease with increasing thickness, and the normalized amplitudes for both emission and transmission almost coincide. The examination of THz transmission and emission, alongside the delay in the emitted pulse, offers evidence that the emitted beam stems from long-range transport in Pt in the case of Ni/Pt.

Finally, to further strengthen our argument we have done the same measurement with negligible SHE material system Ni (3 nm) / Ru (x nm) with $x = 3, 6, 9, 12, 18$ as shown in Figure R3.

As we can see there is a shift in the THz as we increase the thickness of Ru from 3 nm to 6 nm, however for thickness greater than 6 nm there is no observable shift. THz transmission also shows no observable shift. This indicates that the orbital relaxation length to be less than 6 nm. Experimentally it has been calculated that the orbital relaxation length of Ru is around 3.8 nm^7 , thus proving our argument. We have added this in the Supplementary Section S10 in the updated manuscript.

Figure R3: THz measurement of Ni (3 nm) / Ru (x nm) where $x = 3, 6, 9, 12, 18$
 (a) Terahertz emission measurements. The delay in the emitted pulse is absent beyond 6 nm indicating an orbital relaxation length of $<6 \text{ nm}$. (b) Terahertz Transmission measurements.

The modification in the supplementary information is indicated below.

Section S10: THz emission and transmission in Ni (3 nm)/ Ru (x nm)

Figure S9: THz measurement of Ni (3 nm) / Ru (x nm) where $x = 3, 6, 9, 12, 18$ (a) Terahertz emission measurements. The delay in the emitted pulse is absent beyond 6 nm indicating an orbital relaxation length of <6 nm. (b) Terahertz Transmission measurements

Figure S9 shows the THz emission and transmission from Ni (3nm)/ Ru (x nm) with $x = 3, 6, 9, 12, 18$. Previously experimentally it is verified that the orbital diffusion length of Ru is around 3.8 nm. Therefore, as the thickness of Ru is increased from 3 nm to 6 nm, there is a considerable shift in the emitted pulse. However, as we increase the thickness further, the delay is not observable. This also proves that the delay in Ni/Pt (x nm) is because of the long transport entity travelling in Pt.

Figure 2: Data presentation for Ni(3)/Pt(x) where x varies from 3 to 18 nm and NiFe(3)/Pt(x) where x varies from 1 to just 8 nm is misleading and hence the comparisons in the behaviors in subparts (c) and (d) are erroneous.

Response to the Reviewer: We thank the reviewer for the careful review. However, we respectfully highlight that the experimental parameters are not “misleading” but sufficient to make our arguments which we discuss as follows. It has been previously observed that with increase in Pt thickness till 16 nm, there is no delay in the emitted THz pulse from Fe / Pt heterostructure. Please refer to Dr. Tom Seifert’s talk here⁸ and the plot adapted from that talk is shown in Figure R4. Similar results was also observed with Co / Pt heterostructure with Pt thickness till 20 nm⁹. The same argument can also be given for a lower orbital relaxation length material (Ru in our case) as shown in Figure R3 where beyond the orbital relaxation length, there is no shift in the emitted THz pulse till 18 nm.

[Redacted]

Figure R4: Comparison of THz emitted from (a) Fe (3 nm)/ Pt (x nm) when x is varied from 2 nm to 16 nm indicating no shift in the emitted THz pulse with increase in Pt thickness. Figure adapted from Dr. Tom Seifert’s talk. Refer to the weblink.

For the sake of like-for-like comparison that the reviewer has raised, we can compare the shift in the emitted pulse from Ni (3 nm) / Pt (3 nm) to Ni / Pt (9 nm) with NiFe / Pt (4 nm) to NiFe / Pt (10 nm) in Figure R5. The comparison clearly validates that the THz emission from Ni / Pt is because of a transport entity which can go far in Pt.

Figure R5: Comparison of THz emitted from (a) Ni (3 nm)/ Pt (x nm) when x = 3, 6, 9, 18 and (b) NiFe (3 nm)/ Pt (x nm) when x = 4, 6, 8, 10. As seen, no shift in the THz pulse is observed in case of NiFe / Pt (x nm)

For NiFe / Pt (10 nm), no shift is observed, attributed to the shorter spin diffusion length of Pt, experimentally determined to be approximately 1.2 nm - 1.6 nm^{5,6}. However, in the Ni/Pt heterostructure, the presence of a larger orbital relaxation length in Pt gives rise to the delay.

The potential delay stemming from Pt's thickness and refractive index is outlined below.

Any delay in a THz emission system based on spin transport would likely stem from Pt's thickness, given its spin diffusion length in magnetic heterostructures, typically ranging between 1.2 nm and 1.6 nm. To substantiate that the observed delay in emitted THz pulses in Ni/Pt, with increasing Pt thickness, arises from the prolonged relaxation of the orbital current, we have computed the potential delay in the THz pulse attributable to Pt's thickness.

Experimentally it has been measured that the refractive index of Pt in THz, from 0.1 to 3 THz is $n \sim 80$

Extra thickness of Pt THz needs to cover for Ni (3 nm)/ Pt (x nm) where x = 6, 9, 18 w.r.t Ni (3 nm) / Pt (3 nm) = (x-3) nm

$$\text{So, time delay expected} = \frac{(x-3)nm}{c/n}$$

Here c = speed of light in vacuum

For $x = 6$, time delay ~ 0.8 fs

$x = 9$, time delay ~ 1.6 fs

$x = 18$, time delay ~ 4 fs

The same calculation was already added in the Supplementary Section S3 of the original manuscript. We have added the reference 9 in our updated version of the manuscript as reference 46.

The modification in line 207 in the main manuscript is indicated below.

Additionally previous experiment suggests that even when Pt thickness is increased upto 20 nm, no THz pulse shift is observed in case of Co / Pt heterostructure⁴⁶.

To me, the behavior of both the bilayers is the same at this short time scale. Any apparent temporal shift (delay or broadening) in the orders of few fs to 10's of fs looks artificial owing to the fact that FWHM of the femtosecond pulses used to sample the THz pulses is ~ 50 fs or even larger at the sample point in such experiments.

Response to the Reviewer: We appreciate the reviewer's comment. The full width at half maximum (FWHM) of our laser measures approximately 42 fs near the sample position, resulting in a THz pulse width of 1 ps. Our data collection involves multiple data points, with the time difference between them defining our resolution, which stands at about 20 fs.

In Ni / Pt samples, we observe a gradual temporal shift with increasing thickness, surpassing our resolution, contrasting with the random and minimal variation seen in NiFe / Pt samples. To validate this observation, we adjusted our resolution to 120 fs by changing the step size of our delay stage. As anticipated, the inferior resolution led to an inaccurate estimation of the temporal shift, with the pulses overlapping each other as we can see in R6.

Figure R6: THz emission from Ni (3 nm) / Pt (x nm) with $x = 3, 6, 9, 18$ with resolution (a) 20 fs (b) 120 fs. With inferior resolution, the delay in the THz pulse is absent.

We have added the information of the resolution of our measurement in the method section of our manuscript in line 385. The modification is indicated below.

The step size of the delay stage is $3 \mu\text{m}$ which corresponds to a time difference of 20 fs between the data points in THz pulse enabling a precise extraction of the delay in the peak of the emitted THz.

Moreover, such short temporal uncertainties are prone to arise from the variation in the thickness of the substrates used.

Response to the Reviewer: We acknowledge the reviewer’s concern regarding the potential delay attributed to variations in substrate thickness. To address this issue, we employed THz transmission spectroscopy. The results, illustrated in Figure R7, demonstrate that when passing THz through Ni / Pt (x nm) with $x = 3, 6, 9, 18$, there is neither a delay nor pulse broadening in the transmitted THz pulses. This provides conclusive evidence that the thickness variation in the substrates is negligible and has no impact on the emitted THz, thereby ruling out its contribution to the observed delay. The transmission results have been included in the Supplementary Section S5 of the revised version.

Figure R7: Transmitted THz E-Field from Ni (3nm)/Pt (x nm) where $x = 3, 6, 9, 18$ normalized to the transmission in Ni (3 nm) / Pt (3 nm). Neither any shift nor any chirping is observed in the emitted pulse eliminating the effect of the variation of thickness of the substrates.

The modification in the supplementary information is indicated below.

Section S5: Terahertz transmission spectroscopy of Ni (3 nm) / Pt (x nm)

Figure S4: Transmitted THz E-Field from Ni (3nm) Pt (x nm) where $x = 3, 6, 9, 18$ normalized to the transmission in Ni (3 nm) / Pt (3 nm) indicating no delay or chirping in the transmitted pulse.

THz transmission measurement of Ni (3 nm)/ Pt (x nm) with varying x is performed. As seen in Figure S4, the transmitted THz pulses neither have a delay nor a pulse chirping when we pass THz through the samples. This proves that the variation in the thickness of the substrates used is negligible and does not affect the THz emitted and therefore does not contribute to the delay observed. Additionally, this also proves that the delay and chirping seen in the emitted pulse cannot be due to the refractive index of Pt

What are the bidirectional repeatability and backlash error in the motorized delay stage used in the experiments?

Response to the Reviewer: We appreciate the reviewer's insightful suggestion. Our measurement setup involves the utilization of the PI linear delay stage VT-80 (Model no 62309250-0000), having a backlash error of 0 and bidirectional repeatability of 10 microns, as specified in the reference¹⁰. However, we remain attentive to the potential backlash error inherent in the delay stage, prompting us to exclusively conduct monodirectional scanning.

In our approach, we consistently scan the delay stage in a single direction to mitigate any potential backlash issues. The monodirectional repeatability of our delay stage is 0.8 microns¹⁰. This strategy allows us to replicate our emitted THz pulses during repeated scans, as illustrated in Figure R8. Notably, the THz pulses overlap, enabling us to obtain averaged data for enhanced clarity. We have added this information in Supplementary Section S9.

Figure R8: THz emitted from Ni (3 nm)/Pt (3 nm) with 2 forward scans sitting in top of each other.

The modification in the supplementary information Section S9 is indicated below.

Section S9: Backlash error of the delay stage

The measurement setup involves the utilization of the PI linear delay stage VT-80 (Model no 62309250-0000), having a backlash error of 0 and bidirectional repeatability of 10 microns, as specified in the reference³. However, to ensure the potential backlash error inherent in the delay stage, monodirectional scanning is conducted exclusively.

In this approach the delay stage is scanned in a single direction to mitigate any potential backlash issues. The monodirectional repeatability of our delay stage is 0.8 microns⁴. This strategy allows the replication of the emitted THz pulses during repeated scans, as illustrated in Figure S8

Figure S8: THz emitted from Ni (3 nm)/Pt (3 nm) with 2 forward scans indicating negligible effect of the backlash error of the delay stage

How was the laser fluence controlled and whether the delay for different OD's was properly considered?

Response to the Reviewer: We appreciate the reviewer's critical comment regarding our fluence control procedure. The laser fluence was regulated using a linear ND filter. Positioned before the emitter, the linear ND filter was adjusted using a mechanical translation stage to modulate the applied fluence. Upon investigating the reviewer's concern regarding whether the delay arises from changes in optical density of the filter, we determined that this is not the case. To demonstrate, we did follow two tests as shown in Figure R9.

1. We put the same filter in front of a 1 mm ZnTe detector and monitor the delay in the detected THz pulse emitted from a LN crystal. Now by changing the position of the filter, we changed the probe power detecting the emitted pulse. The detected THz signal shows no sign of shift as we change the probe power indicating that no temporal shift is introduced because of the change in optical density of the filter as clearly seen in Figure R9(a).
2. As another test, we put the filter in front of a LN emitter and monitored the possible delay in the emitted THz by changing the pump power. As seen in Figure R9(b), there is no shift in the emitted THz pulse. These two tests help us confirm that the delay introduced due to change in fluence is not due to the change in optical density of the filter.
3. Also, the absence of delay in the NiFe / Pt due to change in fluence (Figure R10) indicates that the delay is not because of the change in Optical Density of the ND filter.

Figure R9: THz recorded from LN emitter by changing the position of the Linear ND Filter (a) When the filter is in front of the detector (b) When the filter is in front of the emitter. Both indicate no shift in the THz pulse eliminating the effect of optical density of the filter.

We have added the fluence control method in the Supplementary Section S7. The modification in the supplementary information is indicated below.

Section S7: Delay due to change in Optical Density of the filter

The laser fluence was controlled by a linear ND filter. Before the emitter a linear ND filter was put, and it was moved by a mechanical translation stage to change the applied fluence.

To ensure that the THz shift is not because of the change in optical density of the ND filter, two experiments were conducted

1. The same filter was put in front of a 1 mm ZnTe detector and the delay in the detected THz pulse emitted from a LN crystal was monitored. By changing the position of the filter, probe power of the detecting the emitted pulse was changed. The detected THz signal shows no sign of shift as we change the probe power indicating that no temporal shift is introduced because of the change in optical density of the filter as clearly seen in Figure S6(a).
2. As another test, the filter was put in front of a LN emitter and the possible delay in the emitted THz by changing the pump power was monitored. As seen in Figure S6(b), there is no shift in the emitted THz pulse.

Figure S6: THz recorded from LN emitter by changing the position of the Linear ND Filter (a) When the filter is in front of the detector (b) When the filter is in front of the emitter Both indicate no shift in the THz pulse eliminating the effect of optical density of the filter

The apparent temporal delays have been attributed to ballistic orbital transport in Ni/Pt as compared to spin transport in NiFe/Pt. However, if the delays are correct and such an inference is to be believed, then it rather suggests the opposite picture, i.e., ballistic spin transport due to no temporal shifts in NiFe/Pt.

Response to the Reviewer: The observed temporal delay in the Ni/Pt system is ascribed to the longer orbital transport relaxation length in Pt, as elucidated in Line 207 of our updated manuscript (Line 204 in the older version). In case of spin transport, the short spin relaxation length of 1.2 nm - 1.6 nm^{5,6} in Pt suggests that any delay in the THz pulse emitted is likely due to the refractive index and thickness of Pt. As detailed in our Supplementary Section S3 and confirmed by THz transmission measurements (Supplementary Section S4), the potential delay resulting from Pt's thickness and refractive index is estimated to be approximately 4-6 fs, in contrast to the 90-fs delay presented in our manuscript. So, no temporal shift in NiFe / Pt suggest that the transport entity governing the THz emission in NiFe / Pt must be short ranged.

The ballistic transport phenomenon can be linked to the linear relationship between the delay and the increase in Pt thickness. As noted by the reviewer, a similar delay has been documented in a Ni/W heterostructure in Seifert et al. (Nature Nanotechnology 18, 1132–1138, 2023)¹. Additionally, we draw attention to another paper in Nature Communications by Xu et al. demonstrating a similar delay in the context of Ni / Cu². We have modified the main manuscript in Line 209 accordingly.

The modification is indicated below.

Therefore, the delay in Ni/Pt could be attributed to a long-distance transport in Pt thus ruling out the spin transport as the spin diffusion length of Pt is shorter (~1.2 nm)^{42,43} proving that the THz emission is due to the orbital transport in Ni / Pt heterostructure.

Figure 3: Fluence dependent data for NiFe/Pt(x) for the apparent temporal shifts in the THz pulses with respect to x should also be presented to make a better comparison with the same for Ni/Pt(x).

Response to the Reviewer: We appreciate the reviewer for their valuable suggestion. It is evident that investigating fluence-dependent temporal shifts in THz pulses will enhance clarity. The corresponding data is illustrated in Figure R10. Notably, there is no observable temporal

shift with variations in fluence for NiFe / Pt (x nm). We have incorporated the fluence data of NiFe / Pt (x) in the supplementary section S6 of the updated manuscript.

Figure R10: Fluence Dependent Study of THz emission from NiFe (3 nm)/ Pt (x nm) when (a) $x = 4$ nm, (b) $x = 6$ nm, (c) $x = 8$ nm, (d) $x = 10$ nm indicating no shift in the emitted pulse with increase in laser fluence

Section S6: Fluence dependent study of NiFe (3 nm)/Pt (x nm)

Figure S5: Fluence Dependent Study of THz emission from NiFe (3 nm)/ Pt (x nm) when (a) $x = 4$ nm, (b) $x = 6$ nm, (c) $x = 8$ nm, (d) $x = 10$ nm indicating no shift in the emitted pulse with increase in laser fluence

In my opinion, the results presented do not match the claims of the paper. The assumption that there is a critical value of the laser fluence to change the nature of the orbital transport, i.e., increasing or decreasing across this point by increasing the fluence, cannot be said unambiguously. The raw data has large experimental errors in the way the temporal shifts can be estimated exactly from them. With the increasing fluence, the THz signal from the spintronic heterostructures is believed to saturate. At the high fluence values used in the current paper, it will have a negative impact on the clear estimation of the temporal shifts.

Response to the Reviewer: We express our gratitude to the reviewer for their insightful evaluation. We acknowledge the potential saturation of spintronic or orbitronic emitters at higher fluences. Based on our data analysis, we observe that the amplitude of the emitted THz continues to rise even at the fluence levels examined in our study. As illustrated in Figure R11, depicting the relationship between Emitted THz amplitude and fluence for Ni (3 nm) / Pt (3 nm), we note a consistent increase in THz amplitude up to the highest fluence tested (1270 $\mu\text{J}/\text{cm}^2$). Moreover, we observe no indication of a plateau in the peak THz, thereby facilitating a dependable method for extracting temporal delays.

Figure R11: Fluence vs emitted THz peak to peak E-Field amplitude for Ni (3 nm)/ Pt (x nm) indicating still increasing THz emission with increase in fluence.

Reviewer #2 (Remarks to the Author):

The authors report on an experiment where they pump nickel and permalloy layers with an adjacent platinum layer of varying thickness with a femtosecond laser of 800 nm wavelength. Afterwards, they collect the Terahertz radiation emitted by this system and probe its time delay. From the time delay measured, they draw conclusions on a new phenomenon, ballistic transport of electron current that carries an orbital momentum.

On the first glance, the paper reads well. The text is written in good English, the experiment is well described, and the measured data is shown. However, when diving deeper into the matter, the paper shows many deficits. I cannot recommend it for a publication in Nature Communications (and potentially also not elsewhere in the current state).

Whenever going away from the measured data, the background information, the discussion and conclusions read very general and superficial. There are a lot of claims wherever the proof or justification is not evident to the reader. I am going to discuss some of these statements in the following.

Starting from the abstract and the introduction, they are written in a very general manner. There are a lot of statements that are being made in a way I would expect clear explanations to the reader that are backed up with sound references. For instance, the authors claim that one can induce orbit currents into light metals using “various orbital pumping techniques”. What these techniques are remains disclosed to the reader; the only reference given is a non-peer-reviewed article on arxiv.org. For such a central point to the study, I would expect both a good explanation why and how this works as well as one – ideally more – sound references

Response to the Reviewer: We thank the reviewer for highlighting our work is written in “good English” and that our experiments are “well-described”. We acknowledge the reviewers concerns on generic statements and we have revised the abstract and introduction as per the reviewer’s suggestions.

Orbital Angular Momentum pumping through magnetization dynamics can be explained by transfer of spin angular momentum to orbital angular momentum through Spin Orbit Coupling (SoC) in ferromagnet¹¹⁻¹³. This can be achieved in two ways. First, through ultrafast photoexcitation of a high SoC ferromagnet (like Ni or Co) which results in generation of spin current. The spin current can be converted to orbital current mediated by spin orbit interaction in the ferromagnet making it an indirect source of orbital current^{1,14}. Second is through lattice

dynamics¹⁵. This phenomenon can be described as the inverse mechanism of crystal field torque¹⁶, wherein an out-of-equilibrium orbital angular momentum (OAM) is induced by external disturbances like an electric field and subsequently absorbed by the lattice. We have added this description in the manuscript with added references.

The modification in line 69 in the updated version of the main manuscript is indicated below.

Orbital Angular Momentum pumping through magnetization dynamics can be explained by transfer of spin angular momentum to orbital angular momentum through Spin Orbit Coupling (SoC) in ferromagnet¹⁹⁻²¹. This can be achieved through ultrafast photoexcitation of a high SoC ferromagnet (like Ni or Co) which results in generation of spin current. The spin current can be converted to orbital current mediated by spin orbit interaction in the ferromagnet making it an indirect source of orbital current. Another technique of orbital pumping is through lattice dynamics via efficient orbital dependent electron-phonon coupling²². This can be explained as a reverse process of crystal field torque²³. The orbital current then transports to the adjacent nonmagnetic metal layer where it converts to an accelerated charge current generating THz waves as shown Fig 1(a)^{24,25}. Recent research has proposed that orbital transport can be ballistic and hence can propagate over long distance reaching up to 80 nm in some of the widely used nonmagnetic layer like tungsten at a velocity measured upto 0.14 nm/fs²⁵. Various other works have also described a long range transport of orbital current in various metals²⁶⁻²⁸ and THz emission²⁹.

Further, the authors claim that spin transport is “super diffusive in nature”, giving one reference with a very specific material system and application case (ultrafast demagnetization). A brief web research yields a wealth of publications both for experimental proof of ballistic spin transport (<https://doi.org/10.1103/PhysRevB.65.155322>, both length of 150 nm and velocity of > 0.2 nm/fs exceeding the value of this publication), theory (<https://doi.org/10.1103/PhysRevB.96.115445>), or studies that even investigate when spin transport gets from a diffuse to ballistic regime

(<https://doi.org/10.1103/PhysRevLett.124.196602>). I may assume that the authors believe that spin transport USUALLY occurs in a diffuse regime for their material system, but the statement here is certainly not correct.

Taking into account that the fundamental method to distinguish spin and orbit transport mechanisms in this study is the probing of the time delay under the assumption that a diffuse transport mechanism does not lead to time delay variations when the platinum layer thickness is varied (by the way, shouldn't ballistic transport be faster than diffuse transport?), the whole central argumentation of the paper breaks down when one knows that spin transport can be ballistic.

Response to the Reviewer: In the emitted THz there are few characteristics that we want to highlight which may not have been clear.

1. The delay in the peak of the THz pulse with increase in thickness of Pt.
2. The delay is following a linear trend with thickness at least till 18 nm.
3. The widening of the THz pulse as we increase the thickness of the nonmagnetic layer.

Observations 1 and 3 imply towards a long-range transport mediated THz emission from Ni/Pt whereas observation 2 indicates the ballistic property of that transport process.

We agree with reviewer's comment that the spin transport can be long range and ballistic in some material cases as pointed in the references. However, all the references that the reviewer mentioned have a different material system other than Pt. It is generally well known and well established that the spin transport in Pt in a magnetic heterostructure is short range and super diffusive in nature^{5,6}. Additionally, the spin diffusion length in Pt has also been calculated experimentally and theoretically previously to be around 1.2 nm^{5,6} in these magnetic heterostructure. Hence, if the THz emitted in Ni/Pt is solely because of spin current, it would not have experienced the delay which we have seen (As verified in NiFe / Pt system in our case and seen in the reference Fe / Pt⁸ and Co / Pt⁹, please see Figure R4 and Dr. Tom Seifert's talk here⁸). Note that, the possible delay because of the spin current mediated THz emission in Pt based magnetic heterostructures can only be due to the thickness of Pt layer which can go as high as 5-6 fs which can be verified by the THz transmission measurement shown in R7. The calculation of that delay is added in the Supplementary Section S3 of the manuscript. Therefore, the transport that governs the THz emission in our system cannot be due to spin transport thus, we conclude that the THz emitted is because of Orbital Transport in Pt.

The reviewer also pointed out that the Ballistic motion should be faster than superdiffusive. We believe the ballistic transport is a relative terminology depending on the distance that the orbital or spin current travel and their relaxation length. It may as well be possible the orbital transport is superdiffusive and do not follow linear trend if we consider thicker nonmagnetic layer resulting a longer distance of travel for the orbital current. We are arguing that the orbital transport in Pt is ballistic for at least 18 nm proved by our measurement, (the thickness till which we have a linear delay and a linear decay in the THz peak). Conversely the spin transport in the references that the reviewer mentioned can also be superdiffusive in a thicker nonmagnetic layer. Therefore, it may not be possible to directly compare velocity of the ballistic and superdiffusive transport and it may vary in different material systems. It has already been shown experimentally that in some of the material systems (Ni/W¹, Ni/Cu²) that the Orbital transport is indeed ballistic with velocity similar to what we have extracted.

We acknowledge the reviewer's feedback and for better clarity have added a few lines about possible ballistic spin transport in the main manuscript.

The modification in line 204 in the updated version of the main manuscript is indicated below.

It is worth noting that the spin transport can be ballistic and long-range order in other material systems and interfaces^{44,45}

Is there any other way in the emission characteristics to determine whether the emitted Terahertz radiation has been created by a spin- or an orbit-transport mechanism? In any case, the orbital momentum must be conserved somehow. It would be possible to probe the emitted beam whether it carries an angular orbital momentum (under the assumption that the mechanism is as coherent as the authors claim in the model, which they describe in Figure 4).

Response to the Reviewer: We thank the reviewer for raising this very important concern. As the magnetization and symmetry dynamics of both spin and orbital transport is similar, it is quite challenging to segregate the two. But THz emission spectroscopy is one of the most viable options for differentiating the orbital and spin transport because of its ability to probe the different temporal dynamics of the two. A comparison of THz emission and transmission spectroscopy of Ni/Pt and NiFe/Pt heterostructures with different Pt thickness is shown in Fig R12.

We expect that, beyond the relaxation length, both THz transmission and THz emission will exhibit similar trends with increasing thickness and will coincide with each other. Figure R12 illustrates a comparison between THz transmission and emission for NiFe / Pt and Ni / Pt. In the case of Ni / Pt (x nm), the peak-to-peak THz pulse emitted linearly decreases, contrasting with the exponentially decreasing transmitted peak-to-peak THz pulse, until they converge at a thickness of 18 nm, indicating a relaxation length of approximately 18 nm. However, for NiFe / Pt, both the transmitted and emitted pulses decrease exponentially with increasing thickness, and the normalized amplitudes for both emission and transmission coincide. The comparison of THz transmission and emission, along with the delay in the emitted pulse, provides evidence that the emitted beam is due to a long-range transport in Pt in the case of Ni/Pt. We have added this explanation in the Supplementary Section S4 in the updated version of manuscript along with a few lines in the main manuscript.

Figure R12: Comparison of THz transmission and THz emission for (a) Ni (3 nm) / Pt (x nm) showing a linear decay in the emitted THz pulse in contrast to an exponential decay in the transmitted THz pulse indicating a longer transport phenomena (b) NiFe (3 nm) / Pt (x nm) showing a similar coinciding exponential decay in both emitted and transmitted THz pulse indicating a shorter transport phenomenon.

The modification in the supplementary information Section S4 is indicated in page 6 of the response letter.

The modification in line 216 in the updated version of the main manuscript is indicated below.

A linear decay of the THz signal with normalised to pump fluence at different thickness of Pt (Supplementary Section S4, Figure S3) gives a strong indication of long-range transport mechanism.

Another uncertainty factor is that other effects must be excluded as the authors correctly reference to in line 138, reference 30. However, reference 30 describes that the ultrafast inverse spin-Hall effect (ISHE) easily dominates the emitted radiation and that the underlying effects must be disentangled with great care. However, I miss a clear and conclusive description how this was done during the experiment.

Response to the Reviewer: We thank the reviewer for raising an important point about our work. Our explanation proves that the Orbital current mediated effect dominates over ISHE (which is basically spin transport mediated effect) in THz emission from Ni/Pt heterostructure. The other methods which can contribute to the THz in these kinds of systems is the THz emitted due to just the ferromagnet Ni film. The odd in signal of the emitted pulse can be easily removed by subtracting the THz signal with same applied magnetic field but in opposite polarity. The even in signal from Ni which arises by Anomalous Hall Effect in Ni is almost negligible in this thickness regime of Ni in comparison to the emitted signal from Ni/Pt heterostructure. We have added this description in the updated version of the manuscript. However, it is an open challenge to distinguish the individual components of the Orbital and Spin transport phenomena towards THz emission in such systems and is an aim of our future research direction.

The modification in the main manuscript in line 152 in the updated version is indicated below.

To isolate magnetic effect, we have considered the signals arising from the difference in the signal when the magnetic field is reversed ($[E_{THz}(+M) - E_{THz}(-M)]/2$). The signal due to anomalous hall effect in Ni is nearly insignificant compared to the emitted signal from Ni/Pt in the thickness regime of the nickel layer investigated.

If we have a look to Figure 2d and the description that the widening of the peak is due to angular dispersion within the thickness, I ask myself whether this is the only possible explanation of a widening of that peak in the time domain. I can imagine a ton of other reasons, like strain, higher defect density or other physical properties that are varied with thickness that may influence the transport properties and widen the emission peak.

Response to the Reviewer: We thank the reviewer for the important concerns in calculation and reasoning of the widening of pulse in the emitted THz pulse. Indeed, it is possible that some other external factor which are varied with thickness may influence the transport properties and widen the pulse. However, we argue that widening of the pulse is a result of long-range transport out of which angular dispersion of the transport entity plays a major role along with the other minor factors listed by the reviewer. For a short-range transport, for example spin transport in Pt, the listed factors such as strain, defect density and other physical properties which vary with thickness will not have enough impact as the transport entity does not have enough distance to travel to get affected by these phenomena. As a result, THz will not be able to detect such changes resulting in absence of widening of pulse even for high thickness Pt in NiFe / Pt. Moreover, the pulse width in the emitted THz pulse is tuned gradually which will not be possible if the tuning arises because of the uncertainty factors mentioned by the reviewer. Additionally, the linear decay in the emitted THz peak-to-peak as shown in Figure R12 (a) also indicates the larger dispersion of the orbital current in Pt.

We also have done the THz transmission measurement (see Figure R7) to show that the THz pulse is not delayed or widened due to the thickness of the Pt during emission of the THz pulse from Ni/Pt.

The modification in line 224 in the updated version of the main manuscript is indicated below.

A gradual chirping of THz pulse also eliminates the additional factors such as strain, higher defect density or other physical properties. Furthermore, the existence of these minor factors will not be reflected in the emitted THz pulse driven by a very short transport phenomena, such as spin transport in Pt with relaxation length of around 1.2 nm^{42,43}. This further confirms that a longer transport mechanism is responsible for THz emission in Ni/Pt heterostructure.

Another statement of the paper that worries me is the distinction of non-localized and localized electrons. I learned in fundamental physics that no electron is localized (says Heisenberg), and that orbitals give a probability of localization. Conduction bands in a metal can be seen as a large degenerate orbital (I'm fine with the description of a non-localized electron in this case), but what is the counterpart? Is it core electrons, or states that are in vicinity of a certain lattice location? The whole explanation remains diffuse at this point.

Response to the Reviewer: We extend our apologies for any confusion stemming from our explanation of spin angular momentum transfer. The reviewer rightfully highlighted the lack of clarity in the explanation of non-localized and localized electrons in our manuscript. As the reviewer correctly pointed out, the non-localized electrons in this case refer to the electrons in the conduction bands in the metal.

Conceptually, three angular momenta govern spin and orbital transport in our system:

1. Angular momentum carried by the conduction electrons which are basically the non-localised electrons in our case, embedded within the spin and orbital components of the wave function.
2. Spin angular momentum, encoded within the local magnetic moment arising from magnetic ordering in the materials.
3. Mechanical angular momentum of the lattice, which can be characterized as the phonon angular momentum.

Spin orbit coupling mediates the angular momentum transfer between spin and orbital degree of freedom. Crystal field potential leads to a transfer of orbital angular momentum exchange between electrons and lattice whereas exchange interaction enables spin transfer between local magnetic moment and electron's spin.

In Figure 4(b) of our manuscript, we tried to elucidate the fundamental mechanism underlying the transfer of above-mentioned angular momenta. An ultrafast photoexcitation triggers the exchange of spin angular momentum between the local magnetic moment and the electron's spin, a phenomenon we refer to as the transfer of angular momentum between localized and non-localized electrons.

Figure R13 [Reference to Figure 4(b) in our manuscript]: Transfer of angular momenta in the solid driven by ultrafast photoexcitation and electric field perturbation of the laser fluence. Nevertheless, we have since refined and revised this statement in the manuscript. The modification in the main manuscript in line 297-298 in the updated version is indicated below.

As we apply laser fluence, the **local magnetic moment** couples with the **spin** of the **nonlocalized conduction electrons** through exchange interaction^{10,23}, thus creating a spin current.

Apart from these fundamental concerns, I also have a couple of minor comments where the paper can be improved in technical details:

1. Figure 2c and d: I would swap the x and y axes.

Response to the Reviewer: We thank the reviewer for the suggestion and have done the required changes.

2. There are no error bars.

Response to the Reviewer: We have added the error bar in our measurements. For Figure 2, the error bars are added in the main manuscript. However, Figure 3, the data with error bars are added in the Supplementary Section S7.

3. Figure 3f: I cannot see the relation that is indicated by the arrow. I can see it in Figure 3e, but certainly not in Figure 3f (look, e.g., at the purple curve).

Response to the Reviewer: We thank the reviewer for raising the concern and we sincerely apologize for the confusion created by our Figure 3f. We have made the necessary correction in the figure.

4. Caption to Figure 3 is incomplete.

Response to the Reviewer: We thank the reviewer for correctly pointing out the confusion. We have made the necessary changes in the caption.

References

1. Seifert, T. S. *et al.* Time-domain observation of ballistic orbital-angular-momentum currents with giant relaxation length in tungsten. *Nat. Nanotechnol.* (2023) doi:10.1038/s41565-023-01470-8.
2. Xu, Y. *et al.* Orbitronics: light-induced orbital currents in Ni studied by terahertz emission experiments. *Nat. Commun.* **15**, 2043 (2024).
3. Matsuyama, T., Hu, C.-M., Grundler, D., Meier, G. & Merkt, U. Ballistic spin transport and spin interference in ferromagnet/InAs(2DES)/ferromagnet devices. *Phys. Rev. B* **65**, 155322 (2002).
4. Borge, J. & Tokatly, I. V. Ballistic spin transport in the presence of interfaces with strong spin-orbit coupling. *Phys. Rev. B* **96**, 115445 (2017).
5. Zhang, W. *et al.* Determination of the Pt spin diffusion length by spin-pumping and spin Hall effect. *Appl. Phys. Lett.* **103**, 242414 (2013).
6. Seifert, T. S. *et al.* Terahertz spectroscopy for all-optical spintronic characterization of the spin-Hall-effect metals Pt, W and Cu₈₀Ir₂₀. *J. Phys. Appl. Phys.* **51**, 364003 (2018).
7. Santos, E. *et al.* Exploring inverse orbital Hall and orbital Rashba effects: unveiling the oxidation states of the Cu surface. (2024) doi:10.48550/ARXIV.2402.00297.
8. *On-Line SPICE-SPIN+X Seminar: Tom Seifert.* (2024).
9. Qiu, H. S. *et al.* Layer thickness dependence of the terahertz emission based on spin current in ferromagnetic heterostructures. *Opt. Express* **26**, 15247 (2018).
10. VT-80 Linear Stage. <https://www.physikinstrumente.com/en/products/linear-stages/stages-with-stepper-dc-brushless-dc-bldc-motors/vt-80-linear-stage-1206300>.
11. Go, D. *et al.* Orbital Pumping by Magnetization Dynamics in Ferromagnets. Preprint at <http://arxiv.org/abs/2309.14817> (2023).

12. Hayashi, H., Go, D., Mokrousov, Y. & Ando, K. Observation of orbital pumping. Preprint at <http://arxiv.org/abs/2304.05266> (2023).
13. Santos, E. *et al.* Inverse Orbital Torque via Spin-Orbital Intertwined States. *Phys. Rev. Appl.* **19**, 014069 (2023).
14. Kumar, S. & Kumar, S. Ultrafast THz probing of nonlocal orbital current in transverse multilayer metallic heterostructures. *Nat. Commun.* **14**, 8185 (2023).
15. Han, S. *et al.* Theory of Orbital Pumping. (2023) doi:10.48550/ARXIV.2311.00362.
16. Go, D. *et al.* Theory of current-induced angular momentum transfer dynamics in spin-orbit coupled systems. *Phys. Rev. Res.* **2**, 033401 (2020).

REVIEWER COMMENTS

Reviewer #2 (Remarks to the Author):

In my first review, I have pointed out several points with admittedly critical statements. The other reviewer was also very critical. The authors reacted and revised the manuscript with additional text, figures, and explanations. They also provided an extensive response letter. I have the impression that none of my comments was completely ignored or fought against. I'm going to try to judge the answers in the following.

Induction of orbital currents into light metals:

The authors have specified this point. They have provided several references (although half of them are ArchiveX references and as such not essentially peer-reviewed). This point is answered to my satisfaction.

Super diffusive spin transport and prove that the transport mechanism is orbital:

I would like to thank the authors for the comprehensive explanation in their answer. They added a rather brief statement; what is more notable is that the whole chapter is now much more specific to the material system which was investigated, together with the other changes that were made. It is note mentioning that it is still an indirect indication for the anticipated transport mechanism. Maybe the attempt to do a direct measurement of the conserved orbital momentum could still be interesting for the future.

Disentangling other effects:

I thank the authors for their explanation. I'm still not fully convinced that there are no other effects, but the indication for the correctness is now stronger.

Broadening of the peak:

Here I am also not completely satisfied with the answer itself. I have given a couple of possible reasons that came to my mind; the problem remains that the attribution of the broadening to different effects (and even if only minor) remains fuzzy. I can agree with the authors that a long-range transport mechanism is the main factor here; however, this is still not a direct proof for orbital transport. I think what is more important is that the authors show for their material system that spin transport has much smaller lengths; so orbital transport is basically identified by exclusion.

All in all, the paper has now improved significantly. In my opinion, it is still a collection of indications rather than direct proof for orbital transport. On the big picture, however, the paper is now more concise and is helps understanding the different transport mechanisms. I also think it is important to enrich the scientific discussion with such findings, especially with all the exclusion criteria given. In this light, the paper can be published.

Response Letter

Title: Observation of Active Ballistic Orbital Transport

Manuscript # NCOMMS-24-01505A-Z, Nature Communications

The authors are thankful to the reviewer for their time and efforts in examining our manuscript. We have carefully evaluated the constructive comments and concerns of the referee and have addressed them. In the text below, the reviewer's comment is followed by our response, highlighted in blue. The modifications in the main manuscript as well as the supplementary information is indicated inside the textbox, highlighted in red.

Reviewer #2 (Remarks to the Author):

In my first review, I have pointed out several points with admittedly critical statements. The other reviewer was also very critical. The authors reacted and revised the manuscript with additional text, figures, and explanations. They also provided an extensive response letter. I have the impression that none of my comments was completely ignored or fought against. I'm going to try to judge the answers in the following.

Induction of orbital currents into light metals:

The authors have specified this point. They have provided several references (although half of them are ArchiveX references and as such not essentially peer-reviewed). This point is answered to my satisfaction.

Response to the Reviewer: We thank the reviewer for their constructive comments which helped strengthen our work and sincerely thank them for the positive feedback.

Super diffusive spin transport and prove that the transport mechanism is orbital:

I would like to thank the authors for the comprehensive explanation in their answer. They added a rather brief statement; what is more notable is that the whole chapter is now much more specific to the material system which was investigated, together with the other changes that were made.

Response to the Reviewer: We thank the reviewer for their careful review and comment on the transport mechanism involved in this system. It indeed helped us remove the possible ambiguity in our claim. We appreciate his/her positive feedback on our work.

It is note mentioning that it is still an indirect indication for the anticipated transport mechanism. Maybe the attempt to do a direct measurement of the conserved orbital momentum could still be interesting for the future.

Response to the Reviewer: We agree to the reviewer that the direct measurement of the conserved orbital momentum is a very interesting work for the future.

Disentangling other effects:

I thank the authors for their explanation. I'm still not fully convinced that there are no other effects, but the indication for the correctness is now stronger.

Response to the Reviewer: We thank the reviewer for his/her valuable suggestion. The clear disentanglement of all the effects is an interesting project which is a separate work and currently an investigation we are carrying out. A detailed study of THz emission and transmission spectroscopy is a viable option which can help disentangle all the effects.

Broadening of the peak:

Here I am also not completely satisfied with the answer itself. I have given a couple of possible reasons that came to my mind; the problem remains that the attribution of the broadening to different effects (and even if only minor) remains fuzzy. I can agree with the authors that a long-range transport mechanism is the main factor here; however, this is still not a direct proof for orbital transport. I think what is more important is that the authors show for their material system that spin transport has much smaller lengths; so orbital transport is basically identified by exclusion.

Response to the Reviewer: We thank the reviewer for the comment and positive feedback. It is indeed possible that some other minor mechanisms may lead to broadening the pulse. However, as we explained, those minor mechanisms can only be detected by THz if it is accompanied by a long-range transport in the thickness regime investigated. For clarification we have added this in the manuscript. It is indeed true that for this material system, spin transport will be of much smaller length, and hence the delay and chirping in emitted pulse indicating a long-range transport thus proving the presence of orbital transport.

We thank the reviewer for their positive feedback.

The modification in line 229 in the updated version of the manuscript is indicated below.

Furthermore, the existence of these minor factors will only be reflected in the emitted THz pulse in the thickness regime investigated if it is accompanied by a long transport phenomenon thus ruling out the short range spin transport in Pt with relaxation length of around 1.2 nm^{41,42}, resulting in absence of pulse chirping as seen in case NiFe/Pt (x nm). This further confirms that a long-range transport mechanism is responsible for THz emission in Ni/Pt heterostructure, excluding the possibility of short-range spin transport and strongly indicating orbital transport.

All in all, the paper has now improved significantly. In my opinion, it is still a collection of indications rather than direct proof for orbital transport. On the big picture, however, the paper is now more concise and it helps understanding the different transport mechanisms. I also think it is important to enrich the scientific discussion with such findings, especially with all the exclusion criteria given. In this light, the paper can be published.

Response to the Reviewer: We thank the reviewer for his/her positive comments and recommending our work for publication. We also thank him/her for highlighting the significance of our work with respect to various transport mechanisms.